# Numerical Simulation of Dry Ice Compaction Process: Comparison of the Mohr–Coulomb Model with the Experimental Results

**DOI:** 10.3390/ma15227932

**Published:** 2022-11-10

**Authors:** Maciej Berdychowski, Jan Górecki, Krzysztof Wałęsa

**Affiliations:** Faculty of Mechanical Engineering, Institute of Machine Design, Poznan University of Technology, 60-965 Poznań, Poland

**Keywords:** Mohr–Coulomb, Drucker–Prager Cap, densification, compression, carbon dioxide (CO_2_), dry ice

## Abstract

How to reduce consumption of energy in manufacturing has become a topical issue nowadays. Certain manufacturing processes are known for being highly energy-intensive and compression of materials belongs to this group. This article presents the simulation of the process of compression of dry ice snow with the use of the Mohr–Coulomb model. Two simulation variants were considered in this research. In the first one, constant input parameters were used and in the second one, the input parameters were variable, depending on the changing density of the compressed material. The experimental data were compared with the predicted values to find that the model using constant input parameters was inferior as regards to the goodness of fit. On the other hand, the model with variable input parameters was less accurate in predicting the maximum compression force acting in the process. The last section of this article deals with simulations performed with the Drucker–Prager Cap and modified Cam-Clay models. Finally, it was concluded that the Mohr–Coulomb model yields a more accurate representation of the compression process while requiring less information on the variation of the material parameters.

## 1. Introduction

The recent publications in the area of mechanical engineering show a growing interest in research efforts dedicated to management of waste materials [1,2] and to efficient use of electricity [3]. Both these issues are directly related to the efficiency of production processes and to the effort to reduce the consumption of raw materials and emissions to the environment [2]. Noteworthy, these issues are particularly relevant to processing of waste materials, the example of which are lignocellulosic materials [3,4], in the case of which obtaining a positive energy balance of the whole recycling cycle is of essence. This balance can be improved by, for example, adjustment of the shape of tools used for compaction [5] or perforation [6] of the material in question.

Computer simulation techniques are commonly used in the process of development of existing or new production technologies. Appropriate mathematical models representing the behavior of the processed materials allow determination of different process variables, including strains. There are models applicable to the elastic region, such as the Hook model [7] or to the plastic region, such as the Cam-Clay model [8,9].

Carbon dioxide (CO_2_) is one of such waste materials and its compression was the subject of the research project reported in this article. It is waste material generated in large quantities during production of ammonia [10] which is stored in liquid form under a pressure of 18 bar [11,12]. It crystallizes in the process of adiabatic expansion to atmospheric pressure. The specific properties of the resulting Crystallized Carbon Dioxide (CCD) include low temperature, i.e., −78.5 °C [13] and sublimation at room temperature [14]. Both these features are considered desirable except that a high sublimation rate is not desired in refrigeration applications [14,15]. Compression of CCD, for example, into pellets, extends the range of application with, for example, dry ice blasting [16].

Ram or crankshaft extruders are the machines typically used to compress CCD [17]. A typical working mechanism of such machines is shown in Figure 1. The process starts with the introduction of dry ice snow (Figure 1, label 2) into the extrusion barrel (Figure 1, label 1). The barrel includes the die (Figure 1, label 3), which during the process is filled up with compressed CCD allowing us to treat the barrel during the initial phase of the work cycle as a dead-end tube. Next, the extrusion ram (Figure 1, label 4) moves, thus decreasing the volume occupied by the loose material, which results in its compression. This process continues until the value of force *F_Z_* is no longer higher than the extrusion resistance of CCD being pushed through the dies.

During the compression phase, the magnitude of the force exerted on the ram *F_Z_* is lower than the force exerted by the compressed material on the dead end of the barrel *F_D_* that is made up by the material filling up the die cavity. This difference is caused by the interaction of the compressed material with the side walls of the barrel and by friction between their surfaces. Internal friction is yet another factor involved in the process, caused by relative movements of the particles. Both these factors result in dissipation of the initial energy, represented by the decrease of *F_Z_* down to *F_D_*, as mentioned in the first chapter of this section. The compression changes the physical parameters of the material being compressed, including Young’s modulus, Poisson’s ratio and the coefficient of friction, which has been confirmed by the experimental research studies on dry ice reported by Biszczanik et al. (2020), Biszczanik et al. (2021) and Talaśka (2018) [18,19,20]. This justifies the research efforts to derive numerical models that would relate the values of the physical parameters of the material to its density *ρ*.

For modeling of plastic deformation during compression, Wilczyński et al. (2021) and Diara et al. (2012) used the Drucker–Prager Cap model (DPC) [5,21]. This model has been successfully used to simulate the densification processes, similarly to the modified Cam-Clay model [22,23]. In one of their previous studies, the authors of this article compared the applicability of these two models for the process of CCD compression. DPC model was found to provide a much more accurate representation of the actual compression curve, as compared to CC. Additionally, the maximum compression force *F_Z_* predicted by the DPC model was closer to the observed value.

The above models are, as mentioned, widely used for simulation of various industrial processes, yet it is the Mohr–Coulomb (MC) model that found application in simulation of the process of compression, as indicated by Brewin et al. (2008) [22]. It was observed that the MC model requires less experimental inputs, as compared to the DPC and CC models [24]. These inputs are limited to determination of the friction angle *β*, dilation angle *ψ*, cohesion yield stress *τ* and absolute plastic deformation in relation to the density of material *ρ* [22,25].

The results of numerical analyses described in this article are part of a larger research project, as part of which the following items have been completed so far:-laboratory tests to determine the relationship between CCD mechanical properties as a function of the material density [19,20],-test set-up to verify the values obtained in numerical prediction simulations [15,17],-analytical and numerical models to simulate the process [9,10].

The following chart (Figure 2) gives the overview of the whole research project.

The models are expected to predict the change of compression force as a function of ram displacement showing a good match with the experimental data. Such models can then be used in studies to optimize the geometry of the extrusion die, for example, with the application of evolutionary computation techniques.

The literature review did not reveal any publications analyzing the applicability of the MC model for simulation of the CCD compression process, so it was not possible to compare the quality of results of the DPC, CC and MC models.

Moreover, the information on comparison of the results yielded by DPC and MC simulations is scarce. Hence the information provided herein fills the gap in the current knowledge, supporting the choice of the most appropriate model for numerical studies of the dry ice compression and extrusion processes. Greaves et al. (2011) and Kaufman et al. (2020) pointed to the importance of research on dry ice in view of the future utilization of the surface of Mars, which is partly covered by crystallized carbon dioxide [26,27]. The authors believe that the output of this research, together with parameters of the numerical model, may turn out to be useful in the simulation of land surveying work on the surface of that planet.

## 2. Materials and Methods

This article presents the simulation of the process of CO_2_ compression with the use of the Mohr–Coulomb (MC) model. The simulation was performed in Abaqus Explicit 2020 and the results were compared with the published experimental data obtained in the process of deriving of the CCD compression curve [19]. The numerical analyses were required formulation of the model parameters related to the properties of CCD, which is described further in this chapter.

### 2.1. Materials

#### 2.1.1. Dry Ice Snow

Solid carbon dioxide is obtained through expansion of liquid carbon dioxide with initial temperature of −18 °C and 20 bar pressure of storage. The phase transition is an adiabatic process, during which the storage pressure is reduced to atmospheric pressure. As a result, the material crystallizes into dry ice snow of 550 kg/m^3^ bulk density [28,29].

The temperature of dry ice in normal ambient conditions is −78.5 °C and in the form of snow, it rapidly sublimates due to the large contact surface area.

#### 2.1.2. Compression

The numerical simulations and empirical results concerned the compression process. The purpose of compression is to reduce the spacing of particles and, in this way, increase the density of the material and decrease the surface area at the solid-gas phase transition interface. External forces are applied in the process, increasing the density from 550 kg/m^3^ to the maximum level of 1650 kg/m^3^. As observed in the literature, at densities below 1000 kg/m^3^, dry ice is not sturdy enough to be transported or stored [26]. The values of physical parameters, such as the Young’s modulus and Poisson’s ratio are close to null, which considerably hinders the Finite Element Method simulation below that density. This being so, this research has been limited to the density range from 1050 to 1650 kg/m^3^.

Sublimation that occurs in normal ambient conditions was not taken into consideration in the performed simulations. This simplification was justified by roughly the same temperature of the working system components and CCD due to continuous contact between their surfaces. With the sublimation rate close to zero, this phenomenon may well be ignored.

#### 2.1.3. Elastoplastic Properties of Dry Ice as a Function of Density

Han et al. (2008) noted that a change of density changes the elasto-plastic properties of the material and thus the forces involved in the process [23]. As the particles get packed closer together and the contact surfaces increase, so do the values of Young’s modulus *E* [19] and Poisson’s ratio *ν* [20]. These changes in the values of *E* and *ν* as a function of density *ρ* have been studied and reported in the literature and the following equations are given to describe them:(1)$$E\left(\rho \right)=1.328\rho -1282.93,$$
(2)$$\nu \left(\rho \right)=-7.1226\times 10^{-2}+\frac{0.58544\;\rho^{8.223076}}{1220.08^{8.223076}+\rho^{8.22308}}.$$

### 2.2. Method and Numerical Model

The numerical studies described further in this article were conducted in Abaqus 2020/Explicit by Dassault Systèmes. In this simulation, the total strain of the material ε is a sum of elastic strain *ε_e_* and plastic *ε_pl_* as per the following equation:(3)$$\varepsilon =\varepsilon_{e}+\varepsilon_{pl}.$$

The strain in the elasto-plastic region was described using a linear elastic model in which *E* and *ν* are calculated with Equations (1) and (2) above. MC model, in turn, was used to represent the plastic deformation.

The Mohr–Coulomb yield criterion assumes that yielding occurs when the shear stress at any point in the material reaches a level at which a linear relationship with normal stress in the same cross-section can be derived. According to study [25], the material model is based on the Mohr’s stress circles drawn using the differences of appropriate principal stresses. Yielding of the material is represented by a line tangent to both circles (Figure 3).

Thus, the MC model can be defined as follows:(4)τ=c−σtanϕ.
where *σ* is negative when dealing with compression. The following equations are, in turn, derived from the Mohr’s circles:(5)τ=scosϕ,
(6)$$\sigma =\sigma_{m}+s\;\sin\phi .$$

Substituting Equations (4) and (5) into Equation (6), we can represent the MC model as follows:(7)$$s+\sigma_{m}\sin\phi -c\;\cos\phi =0,$$
where: *ϕ* is the internal friction angle, *c* is the coefficient of cohesion, *s* is the maximum shear stress defined as:(8)$$s=\frac{1}{2}\left(\sigma_{1}-\sigma_{3}\right),$$
while $\sigma_{m}$ is the average strain defined as:(9)$$\sigma_{m}=\frac{1}{2}\left(\sigma_{1}+\sigma_{3}\right).$$

The Mohr–Coulomb model was used to derive a numerical model representing the process of compression of bulk material by a compression ram inside a 30 cm diameter barrel. This allowed verification of the output of the numerical calculations with the previously obtained data, including the data from previous experiments. The numerical model used in this study represented in Figure 4 below was used together with the DPC and CC models in an earlier study reported in [9] by the same authors. The numerical model in question is made up of four parts: compressed material (Figure 4, label 3), compression barrel (Figure 4, label 1), dead-end disc (Figure 4, label 4) and upper disc representing the ram (Figure 4, label 2). The only deformable part in this model was the material being compressed. It was represented by a cylinder with diameter *D_C_* of 30 mm and height *h_C_* of 39.95 mm. The other elements of the model representing the compression barrel (Figure 4, label 1) were modeled as discrete rigid surfaces. A discrete rigid surface is assumed to be rigid and is used in contact analyses to model surfaces that cannot deform. The ram (Figure 4, label 2) was modeled similarly to the barrel (Figure 4, label 1). It is represented by a flat disc in the model. The same shape was used to represent the dead end of the compression barrel (Figure 4, label 4).

The barrel and the dead-end disc were modeled as rigid parts with no degrees of freedom. The ram had one degree of freedom of linear movement along Z axis. In the simulation, the ram was traveling along the X axis at a speed of 5 mm/s.

Surface-to-surface contact was defined to represent the interaction between the compressed material and the barrel. Three surface-to-surface contacts were defined for the compressed material: with the inside surface of the barrel, with the ram and with the dead-end disc. Friction coefficient of *µ* = 0.1 [18] was defined as a contact property. Then the properties of the compressed material were defined, based on the experimental data reported by Biszczanik et al. (2021 and 2022) [19,20]. The results presented in this section confirm that, as mentioned above, the properties change as the material is compressed. In order to achieve the best consistency between the predicted and observed data, Abaqus Subroutines VUSDFLD was used to reflect the above-mentioned changes with PEEQ values (Table 1 and Table 2) used as the criterion determining the changes to the material properties. This means that the input values of the parameters defining the parameters used by Abaqus for each calculation step will depend on the equivalent plastic deformation values (PEEQ) obtained in the preceding step.

A measurement point was defined on the inside surface of ram on its symmetry axis. It was used to record and view the calculated reactions induced on the ram by the process of compression and to measure displacements. This definition provides the best representation of the observed forces and displacements.

## 3. Results and Discussion

The simulation results were used to draw the curve representing the change of compression force as a function of ram displacement *s*. The graph below (Figure 5), in turn, compares the predicted and the observed data.

Sum of squared errors (*SSE*) was used to assess the consistency and the fit of the experimental curve to the simulation output data. SSE proved to be useful in comparing the predicted data obtained with different numerical models and the experimental data, as reported in [30,31]. Berdychowski et al. (2022) [9] observed that the value of *SSE* is based on the sum of squared differences between the predicted data *F^S^* obtained from the simulation and the experimental data *F^E^* and gave the following equation:(10)$$SSE=\sum_{s=86}^{100}{\left(F_{s}^{S}-F_{s}^{E}\right)}^{2}.$$

The value of *SSE* was calculated in this case for the range of *s* between 86 and 100 mm, summing the squared differences at 0.1 mm increments. In addition, *SSE* was determined for the values of *s* at 1 mm increments. These parameters are in correspondence with the results of previous studies, allowing a comparison of the MC model with the Drucker–Prager Cap and modified Cam-Clay models in terms of the goodness of fit. The change of *SSE* was represented by the bar diagram in Figure 6 and in Table 3. MC1 index designates the predicated data obtained at constant values of the model parameters while MC2 designates the values obtained when the parameters changed as a function of PEEQ.

The numerical simulation with the use of the Mohr–Coulomb, performed as part of this study, gave an approximate representation of the change of the compression force as a function of ram displacement. Comparing the predicted and observed values, we note an over two times lower value of *SSE* summed over the range from 86 to 100 mm in the case of the model which used variable input parameters. The model using constant input parameters, in turn, featured a lower value of *SSE* for 1 mm intervals between 86 and 90 mm, i.e., in the initial phase of the simulation. On the other hand, in the range of 90–100 mm, a better fit of the observed data was obtained with the model that used variable input data.

Noteworthy, the maximum compression force was higher than the observed value in both simulations and the difference between the maximum values of *F_S_^MC2^* and *F_E_* was almost twice the same difference between *F_S_^MC1^* and *F_E_*.

The above observations allow us to conclude that:-the Mohr–Coulomb model using constant input parameters gives an accurate prediction of the maximum force acting during compression of dry ice,-variable input parameters, depending on the value of PEEQ would, however, be more appropriate if it is required to determine the change of the applied force during the compression process.

As mentioned, DPC and CC models were verified by comparing the predictions with the experimental data [9]. The same method was used and thus it was possible to compare the values of *SSE* of the MC1 and MC2 simulations with the results obtained with DPC and CC models. Table 4 and Figure 7 are given below for this purpose.

The result show that the goodness of fit of the predictions given by the respective models with the experimental data varied depending on the section. In the case of CC model, the lowest value of *SSE* was obtained in the range of 86–88 mm. In the ranges 88–97 mm and 99–100 mm, the lowest *SSE* value was obtained for the DPC model. In the range of 97–99 mm, in turn, the lowest *SSE* value was obtained for the predictions obtained with the MC2 model.

In the range of 88–100 mm, the lowest accumulated value of *SSE* was obtained for DPC model, followed by MC2.

Table 5 compares the maximum compression forces *F_Z_* predicted by models MC1 and MC2 with the values predicted by the models DPC and CC.

The percentage difference κ between a given value and the force limiting value $F_{E}$ obtained during the empirical tests was additionally determined.

## 4. Conclusions

As mentioned in the previous description and in the literature, the MC does not require as much detail on the physical properties of the material as do the DPC and CC [25] models.

The values given in Table 4 and Table 5 for the MC1 (Mohr–Coulomb model using constant input parameters), MC2 (Mohr–Coulomb model in which the input parameters change with the changing density of the compressed material), Drucker–Prager Cap DPC and modified Cam-Clay CC models showed that the MC2 models offer a better match of the predictions to experimental data than the CC models.

A comparison of the *SSE* value over the whole test range and of the maximum compression force *F_Z_* show a better fit to the experimental curve of the predictions obtained with the DPC model, as compared to the MC2 model proposed by the authors in this article.

However, MC may be preferred over DPC in the initial phase of research owing to less detailed input information on the change of mechanical properties as a function of the material density. Nevertheless, DPC should still be the model of choice in the final stage of research.

## Figures and Tables

**Figure 1 materials-15-07932-f001:**
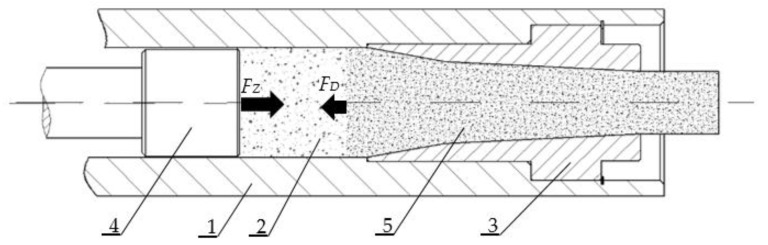
Working system of a ram-type extruder: 1. Compression barrel; 2. Loose material; 3. Die; 4. Extrusion ram; 5. Compressed material [9].

**Figure 2 materials-15-07932-f002:**
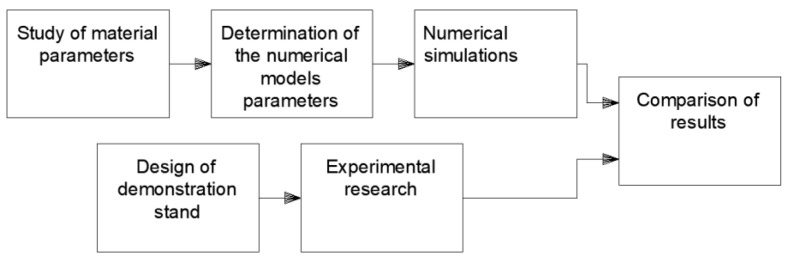
Stages of the overall research project.

**Figure 3 materials-15-07932-f003:**
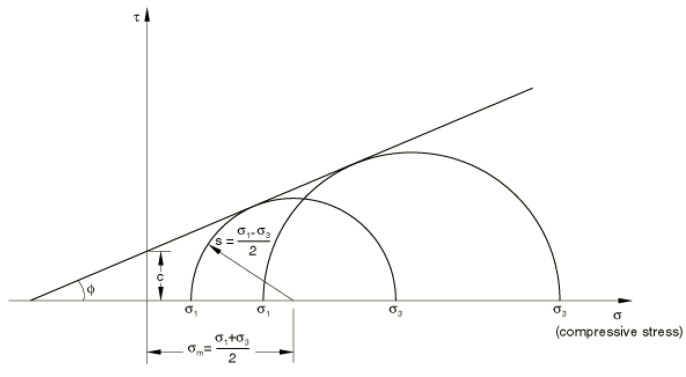
Mohr–Coulomb yield model [25].

**Figure 4 materials-15-07932-f004:**
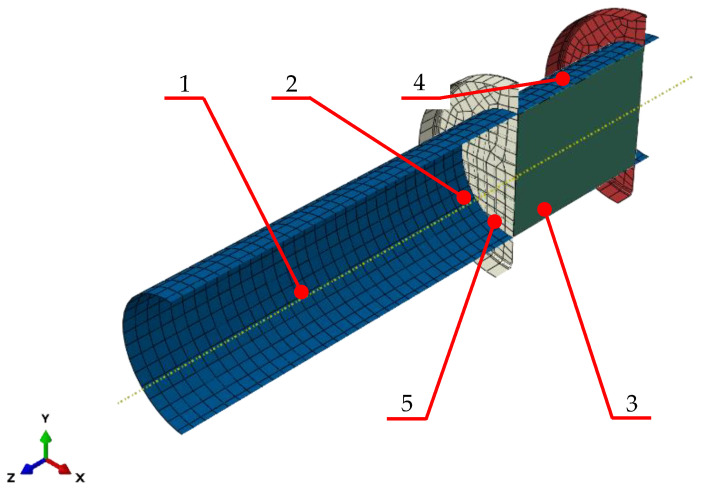
Numerical model: 1 barrel, 2 ram, 3 compressed dry ice, 4 dead-end disc, 5 compression force measurement point [9].

**Figure 5 materials-15-07932-f005:**
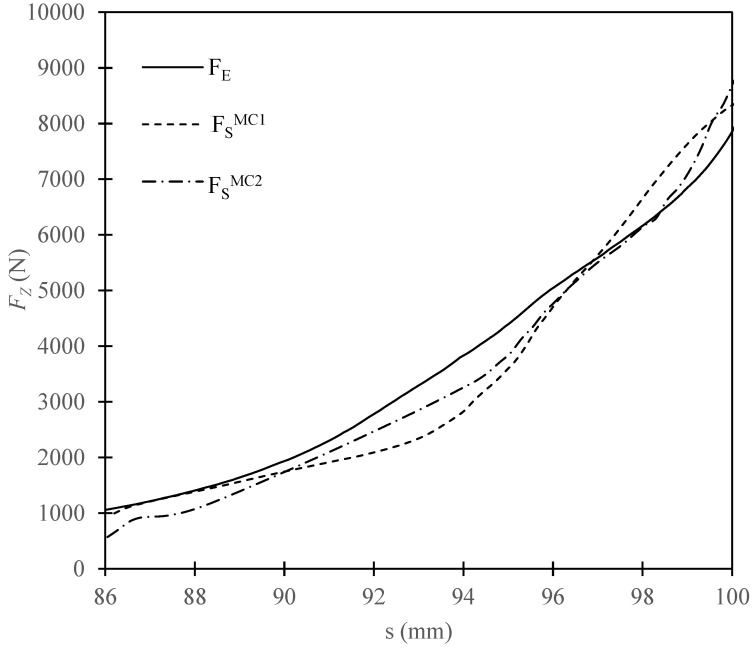
Change of *F_Z_* as a function of *s* for the observed and simulation output data.

**Figure 6 materials-15-07932-f006:**
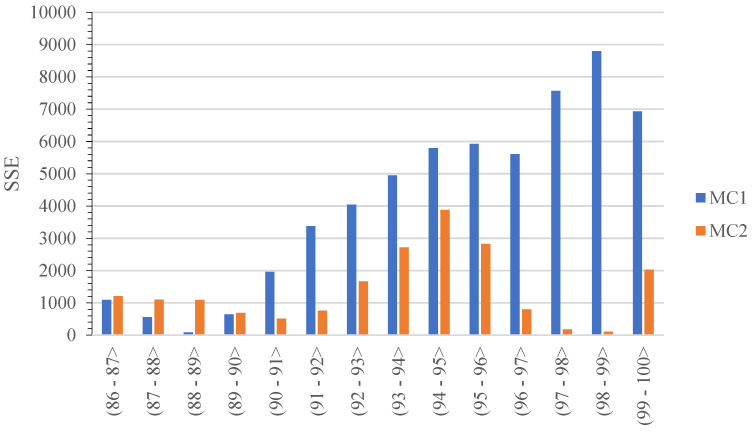
Change in the *SSE* value of MC1 and MC2, in 1 mm intervals.

**Figure 7 materials-15-07932-f007:**
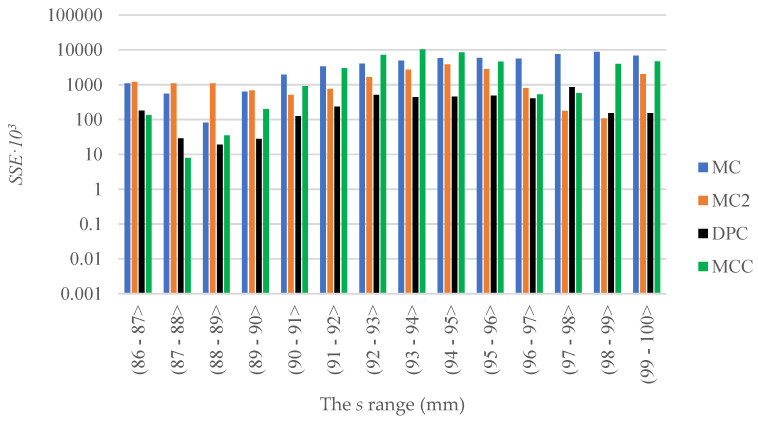
Change in the *SSE* value of MC1, MC2, DPC and CC in 1 mm intervals.

**Table 1 materials-15-07932-t001:** Input data of the simulation using the MC model.

Cohesion Yield Stress	Abs Plastic Deformation	Friction Angle	Dilation Angle
500,000	0	15	0
600,000	0.4		
700,000	0.8		
1,000,000	1		

**Table 2 materials-15-07932-t002:** MC model input data depending on the value of PEEQ.

Young’s Modulus	Poisson’s Ratio	PEEQ
36,590,000	0.02	0–0.5
45,142,000	0.05	0.5–0.9
60,900,000	0.07	0.9–1.5
152,100,000	0.4	1.5–2
823,820,000	0.46	2–5

**Table 3 materials-15-07932-t003:** Values of *SSE* for constant and varying parameters of the model.

Range of *s,* mm	SSE^MC1^	SSE^MC2^
(86–87〉	1.1 × 10^6^	1.2 × 10^6^
(87–88〉	5.6 × 10^5^	1.1 × 10^6^
(88–89〉	8.2 × 10^4^	1.1 × 10^6^
(89–90〉	6.4 × 10^5^	6.9 × 10^5^
(90–91〉	2.0 × 10^6^	5.1 × 10^5^
(91–92〉	3.4 × 10^6^	7.6 × 10^5^
(92–93〉	4.0 × 10^6^	1.7 × 10^6^
(93–94〉	5.0 × 10^6^	2.7 × 10^6^
(94–95〉	5.8 × 10^6^	3.9 × 10^6^
(95–96〉	5.9 × 10^6^	2.8 × 10^6^
(96–97〉	5.6 × 10^6^	8.0 × 10^5^
(97–98〉	7.6 × 10^6^	1.8 × 10^5^
(98–99〉	8.8 × 10^6^	1.1 × 10^5^
(99–100〉	6.9 × 10^6^	2.0 × 10^6^
**(86–100** 〉	**5.7 × 10^7^**	**2.0 × 10^7^**

**Table 4 materials-15-07932-t004:** Values of *SSE* for constant and varying parameters of the models MC1, MC2, DPC and CC.

Range of *s,* mm	*SSE* ^MC1^	*SSE* ^MC2^	*SSE* ^DPC^	*SSE* ^CC^
(86–87〉	1.1 × 10^6^	1.2 × 10^6^	1.34 × 10^5^	1.81 × 10^5^
(87–88〉	5.6 × 10^5^	1.1 × 10^6^	8 × 10^3^	2.9 × 10^4^
(88–89〉	8.2 × 10^4^	1.1 × 10^6^	3.5 × 10^4^	1.9 × 10^4^
(89–90〉	6.4 × 10^5^	6.9 × 10^5^	2.01 × 10^5^	2.8 × 10^4^
(90–91〉	2.0 × 10^6^	5.1 × 10^5^	9.21 × 10^5^	1.27 × 10^5^
(91–92〉	3.4 × 10^6^	7.6 × 10^5^	3.027 × 10^6^	2.38 × 10^5^
(92–93〉	4.0 × 10^6^	1.7 × 10^6^	7.268 × 10^6^	5.16 × 10^5^
(93–94〉	5.0 × 10^6^	2.7 × 10^6^	1.0334 × 10^7^	4.42 × 10^5^
(94–95〉	5.8 × 10^6^	3.9 × 10^6^	8.474 × 10^6^	4.56 × 10^5^
(95–96〉	5.9 × 10^6^	2.8 × 10^6^	4.660 × 10^6^	4.85 × 10^5^
(96–97〉	5.6 × 10^6^	8.0 × 10^5^	5.31 × 10^5^	4.07 × 10^5^
(97–98〉	7.6 × 10^6^	1.8 × 10^5^	5.73 × 10^5^	8.5 × 10^5^
(98–99〉	8.8 × 10^6^	1.1 × 10^5^	3.962 × 10^6^	1.53 × 10^5^
(99–100〉	6.9 × 10^6^	2.0 × 10^6^	4.705 × 10^6^	1.54 × 10^5^
**(86–100** 〉	**5.7 × 10^7^**	**2.0 × 10^7^**	**4.09 × 10^6^**	**4.48 × 10^7^**

**Table 5 materials-15-07932-t005:** The maximum *F_z_* value.

	$!\max F_{Z}\;(\mathrm{kN})$	κ (%)
$F_{S}^{MC1}$	8.397	6.37
$F_{S}^{MC2}$	8.340	5.65
$F_{S}^{DPC}$	8.064	2.15
$F_{S}^{MCC}$	8.328	5.50
$F_{E}$	7.894	

## Data Availability

Not applicable.

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
