# Peer review of "Numerical Simulation of Dry Ice Compaction Process: Comparison of the Mohr–Coulomb Model with the Experimental Results"

_materials, 2022, doi:10.3390/ma15227932_

Round 1
Reviewer 1 Report
In this study, a simulation of the process of compression of dry ice snow with the use of the Mohr-Coulomb model was developed. The experimental data were compared with the predicted values. Besides, two models, i.e. Drucker Prager/Cap and modified Cam-Clay models were used and their performance were compared.
There are some concerns from the reviewer.
1. The background of the compression of dry ice snow might be highlighted. As well as the drawbacks or challenges of numerical model to simulate the dry ice extrusion. In the introduction part, the authors talk a lot on the energy, emissions that are not strong related to the topic.
2. In the introduction part for literature review, the authors should address the difference of this manuscript to the published work of the same group, to name a few, the reference 9,10,15,17,19,20,24 in this manuscript. Especially reference 9.
3. As a full-length article, the results and discussion part in this manuscript is too short. Only one figure and one table were presented. A more detailed description of model, discussion, and clarification of model performance might be given.
4. Show the full name of abbreviations when it appears first time, for example DPC. Could it be Drucker Prager/Cap model? As well for FEM.
5. Some errors:
Label 1 to 5 are missing in figure 1.
In line 143 and line 155, there are two equation (3). Should be revised.
Journal name of ref 28 is missing.
Author Response
Review 1
Remarks:
- The background of the compression of dry ice snow might be highlighted. As well as the drawbacks or challenges of numerical model to simulate the dry ice extrusion. In the introduction part, the authors talk a lot on the energy, emissions that are not strong related to the topic.
Response:
Thank you for your suggestion, the introduction has been expanded with a paragraph highlighting the importance of research on compression and mechanical treatment of dry ice. As we can read in the litatrature, the future colonisation of Mars is just one example of potential applications of this research. However, the main objective of the research on the dry ice densification is, as yet, improvement of the quality of the produced dry ice pellets by whic we mean a higher density and less electricity used in production. That is why we speak so much about energy consumption in the introduction.
- In the introduction part for literature review, the authors should address the difference of this manuscript to the published work of the same group, to name a few, the reference 9,10,15,17,19,20,24 in this manuscript. Especially reference 9.
Response:
Thank you for your suggestion, we have included in the intrduction the information on interrelations between the referenced articles of ours. The numerical models presented in the article and in the article referenced in bibliography item 9 are the final step of the project to develop a methodology for designing single-hole dies for compression and extrusion of dry ice.
Finally we refer the interested reader to the earlier research. We believe that we have applied in our research the principles of good practice that can be transferred to other research and development works on other particulate materials. That said, we have added a flow diagram of our research and development works presenting the sequence of stages and specifying the stage which the presented research output relates to.
- As a full-length article, the results and discussion part in this manuscript is too sho Only one figure and one table were presented. A more detailed description of model, discussion, and clarification of model performance might be given.
Response:
Thank you for your suggestion, but please note that that this article describes a part of our study on numerical representation of the process of compression of dry ice snow. With some results published in our earlier articles, we have now focused on verification of the applicability of the MC model, which has been successfully used to describe processing of bulk media, such as soil. We have undertaken to verify if we could use the said model to represent the process of compression of dry ice snow. To this end we have compared the predicted and measured values. SSE method was used to determine how well do the predicted values fit the experimental data. Since the model is described in more detail in section 2 it was left out in section 3. Still, as suggested, we have added a “discussion” part to section 3.
- Show the full name of abbreviations when it appears first time, for example DPC. Could it be Drucker Prager/Cap model? As well for FEM.
Response:
Thank you for your comment. We admit we forgot to explain the acronyms used in the article. Now we have explained the meaning of DPC in line 76 and replaced FEM with the long-form.
- Some errors:
- Label 1 to 5 are missing in figure 1.
Response:
Thank you for your note, we have added the missing labels.
- In line 143 and line 155, there are two equation (3). Should be revised.
Response:
Thank you for your note, the numbering of equations has been corrected starting from equation (4).

Reviewer 2 Report
The reviewer respects the challenge of the authors of this manuscript.
The reviewer expects innovative applications of this MC model in practice.
On the other hand, as an academic journal, this manuscript must respond with revisions to the following comments by the reviewer.
(1) Currently, most of Section 1 is occupied by reviews of previous studies. The authors must divide the research into pure background, Section 1, which advocates the need for this study, and Section 2, which clarifies the position of this study through a review of previous studies.
(2) In Section 2, there is no description or explanation regarding "methods". Authors should clearly state the "methods".
(3) Section 3 "results" and Section 4 "discussion" should be combined into one section.
(4) The reliability of applying the MC model to practice should be explained. At present, the authors are only making comparisons with experiments.
(5) The description of "conclusions" in Section 4 is inappropriate. The authors should draw more general conclusions.
(6) The final paragraph in Section 4 is just the authors' wishful thinking and is inappropriate for publication in an academic journal.
Author Response
Review 2
Remarks:
- Currently, most of Section 1 is occupied by reviews of previous studies. The authors must divide the research into pure background, Section 1, which advocates the need for this study, and Section 2, which clarifies the position of this study through a review of previous studies.
Response:
Thank you for your suggestion. The information given in section 1 is directly related to a long-term research and development project that is currently nearing completion. We have added a paragraph and a chart to illustrate the current stage against the project framework and relate the respective stages to the previous articles. We pointed out in this paragraph that no mention has been found of the application of the MC model for modelling the process of compression of dry ice snow.
- In Section 2, there is no description or explanation regarding "methods". Authors should clearly state the "methods".
Response:
Thank you for your suggestion. We fully agree this item needs clarification. The method is described in 2.2. We have made appropriate revisions highlighting this part. Now it describes the numerical method that we used to obtain the results given in the article. The description covers the model of the analysed material, design of the numerical model and boundary conditions applied in the simulation.
- Section 3 "results" and Section 4 "discussion" should be combined into one section.
Response:
Thank you for your suggestion, we have revised the article accordingly.
- The reliability of applying the MC model to practice should be explained. At present, the authors are only making comparisons with experiments.
Response:
Thank you for your suggestion. Our purpose was to verify the applicability of the MC model for simulating dry ice compression. Suitability of some other models, such as DPC, was confirmed by the outcomes of earlier experimental studies. This manuscript, however, deals with verification of the suitability of the MC model, which has not been done before. In addition, we compared our results with other studies reported in the available literature to assess the goodness of fit of the MC model in comparison to the DPC and MCC models.
The MC model is extensively described in the literature and has been applied in practice for simulating various processes in the area of soil mechanics, including compaction.
Currently, considering a better match with the experimental curve offered by the DPC model there is no room for practical application of the MC model. That said, we believe the MC model could be successfully used for rough simulations of the process in question, for example as part of the initial stage of research and development projects.
- The description of "conclusions" in Section 4 is inappropriate. The authors should draw more general conclusions.
Response:
Thank you for your note. We have revised the last section accordingly.
- The final paragraph in Section 4 is just the authors' wishful thinking and is inappropriate for publication in an academic journal.
Response:
Thank you for your note, the final paragraph of Section 4 has been removed.

Round 2
Reviewer 1 Report
Although the authors have made some revisions. I want to re-emphasize that the results part in this manuscript is too short. After revision, only one figure and one table were presented. A more detailed description of results from more perspective and discussion on parameters should be given. It is too short to be a full-length article, the authors may consider this manuscript to be a technical note, or short communication.
Author Response
Remarks:
Although the authors have made some revisions. I want to re-emphasize that the results part in this manuscript is too short. After revision, only one figure and one table were presented. A more detailed description of results from more perspective and discussion on parameters should be given. It is too short to be a full-length article, the authors may consider this manuscript to be a technical note, or short communication.
Response:
We appreciate this follow-up of yours. Following your suggestion, we have prepared and included in our manuscript a more detailed discussion of results that includes a comparison with previously reported studies. We believe that in the current shape this analysis will support the choice of the optimum model for the specific numerical studies on dry ice snow or any other similar material.
Reviewer 2 Report
The reviewer understood that the revised manuscript accurately and politely responded to the reviewers' comments.
Author Response
Thank you for your review, we will also try to include your suggestions in future articles.